# Real-Time Elastography versus Shear Wave Elastography on Evaluating the Timely Radiofrequency Ablation Effect of Rabbit Liver: A Preliminary Experimental Study

**DOI:** 10.3390/diagnostics13061145

**Published:** 2023-03-16

**Authors:** Li Shi, Xiaoju Li, Wei Liao, Wenxin Wu, Ming Xu

**Affiliations:** 1Department of Medical Ultrasonics, Hainan General Hospital, Haikou 570311, China; 2Department of Medical Ultrasonics, The First Affiliated Hospital, Institute of Diagnostic and Interventional Ultrasound, Sun Yat-sen University, Guangzhou 510080, China

**Keywords:** real-time elastography, shear wave elastography, radiofrequency ablation, liver, contrast-enhanced ultrasound

## Abstract

Purpose: to evaluate and monitor the timely thermal ablation changes of rabbit liver by using two elastographic methods—real-time elastography (RTE) and shear wave elastography (SWE)—as compared to contrast-enhanced ultrasound (CEUS) and physical specimens. Materials and Methods: 20 ablation zones were created in the livers of 20 rabbits using radiofrequency ablation (RFA). After the ablation, RTE and SWE were used to measure the elastic properties of the twenty ablation zones. The consistency of efficacy evaluation for RTE and SWE measurements was analyzed using the Bland–Altman test. The areas of the thermal ablation zones were also measured and compared according to the images provided by RTE, SWE, CEUS, and gross physical specimen measurement. Results: RTE and SWE could clearly display the shape of RFA ablation zones within one hour after the ablation. The average elasticity ratio for the ablation zone measured by RTE was 3.41 ± 0.67 (2.23–4.76); the average elasticity value measured by SWE was 50.7 ± 11.3 kPa (33.2–70.4 kPa). The mean areas of the ablation zones measured with RTE, SWE, gross specimen, and CEUS were 1.089 ± 0.199 cm^2^, 1.059 ± 0.201 cm^2^, 1.081 ± 0.201 cm^2^, and 3.091 ± 0.591 cm^2^, respectively. The Bland–Altman test showed that RTE and SWE have great consistency. Area measurements by CEUS were significantly larger than those of the other three methods (*p* < 0.05). Conclusion: RTE and SWE are both able to accurately confirm the range of ablation zones shortly after the ablation for rabbit livers.

## 1. Introduction

Localized thermal ablation has become a curative therapy for early hepatocellular carcinoma (HCC), and radiofrequency ablation (RFA) is the most commonly used thermal ablation method [1]. Currently, the main imaging techniques for evaluating the treatment effect after RFA are contrast-enhanced ultrasound (CEUS) and contrast-enhanced CT/MRI (CECT/MRI) examinations [2,3]. During the treatment, clinicians usually need to judge whether the ablation is successful immediately after the ablation, in case they are required to perform supplemental ablation therapy. But CEUS and CECT/MRI have some limitations in timely assessment. For CEUS, the vaporization zone and peripheral hyperemia caused by ablation could affect accurate judgment. For CECT/MRI, the inconvenience of patient transport or expensive costs become a huge challenge. Thus, finding a relatively simple and timely method of judging the ablation effect has become an urgent problem to be solved.

Elastography ultrasound is the application of measuring the tissue strain to assess the hardness of the tissue [4,5]. It is applied as the auxiliary diagnostic tool for thyroid, breast, liver, and other organ lesions [6,7,8,9]. After the ablation, the tissue will generate degeneration and coagulation, and this part will become harder compared with the surrounding normal tissue. Thus, this makes it possible to evaluate the success of the ablation and the edge of the ablation zone [10,11,12]. There are two kinds of elastic methods commonly used in clinics: real-time elastography (RTE) and shear wave elastography (SWE) [13]. There are few reports on assessing and comparing the ablation zone using these two methods simultaneously.

The purpose of this study was to evaluate the timely radiofrequency ablation effect on rabbit livers using RTE and SWE, and compare the ablation areas which were calculated by RTE, SWE, CEUS, and gross specimens.

## 2. Materials and Methods

### 2.1. Materials

#### Animals

This study was approved by the Sun Yat-sen University Animal Care and Use Committee and conformed to the Guide for the Care and Use of Laboratory animals. A total of twenty New Zealand White rabbits of either gender, with body weights of 2.5–3.0 kg, were enrolled in the study. The rabbits were provided by Guangdong experimental animal center.

### 2.2. Methods

#### 2.2.1. Preparation before Operation

The rabbits were fasted for 24 h and then anesthetized with 3% sodium pentobarbital (1 mL/kg) through the ear vein. After anesthesia, the hair on the backs and bellies of the rabbits was shaved, and the electrode plates were attached to the backs of the rabbits. Then, we fixed the rabbits to the table in the supine position.

#### 2.2.2. Ablation Progress

Laparotomy was performed on the animals and the liver was exposed. RFA treatments were performed using a unipolar internally cooled radiofrequency probe (Valleylab, Boulder, CO, USA) with an active tip length of 10 mm. The probe was inserted in the liver for 3 cm, and the tip was 1 cm away from the liver surface (Figure 1). For RFA, the ablation time is 180 s at 30-watt power.

#### 2.2.3. Ultrasound Examination

##### RTE

ESAOTE MyLab^TM^Twice was used to measure the RTE. After identification of the target lesion, the transducer was kept in a stable position without pressure, perpendicularly, to better minimize the compression artifact, and RTE mode was implemented over the conventional ultrasound image. The operator vibrates the probe, and the vibration frequency is kept at 2 times per second. The pressure index is controlled at 3–4. A color signal box of appropriate size was displayed as a colored area, with softer tissue presented as green or red and the harder presented as blue. When the images are stable, the operator freezes the image and selects the satisfactory image for analysis. The elastic ratio of the ablation area and normal liver tissue was measured three times and the average of the measured values was taken. The areas of the ablation zone were calculated according to the dimension of the ablation zone. The standard for satisfactory images is as follows: (1) the pressure is appropriate, and the images are clear and stable; (2) the liver tissue surrounding the ablation zone indicated uniform yellow-green color. The area of the ablation zone was calculated, according to the elastic images (Figure 2).

##### SWE

After that, slight repetitive freehand compression applied on another transducer (Aixplorer, SuperSonic Imagine, Aix en Provence, France) was induced by the operator to obtain an SWE elastogram. A color signal box of appropriate size was displayed as a colored area, where softer was presented as blue and harder was presented as red. When the image was stable, without dot artifacts, it was frozen. Qualitative methods were used to determine image stability. On the elastograms, we used an adjustable color scale set at 90 kPa, which is mostly used in human examination. Elastographic quantitative measurement using a 2 mm region of interest (ROI) (QboxTM; Super Sonic Imagine) placed different regions of the ablation zone. The qualified images indicate that (1) almost all of the color signal box is filled with the colors, which is stable, (2) normal liver indicates a uniform blue, and (3) there are no obvious oppression artifacts. Similarly, the area of the ablation zone was calculated, according to the elastic images (Figure 2).

##### CEUS Examination

CEUS examination was performed using a multi-frequency probe (MyLab^TM^Twice) after a bolus injection of second-generation ultrasound contrast agent (0.5 mL/Kg) consisting of microbubbles filled with sulfur hexafluoride (SonoVue, Bracco, Milan, Italy). The conditions of the ablation zone were observed, and the dynamic images were stored. The area of the ablation zone was calculated, according to the CEUS images.

#### 2.2.4. Animal Euthanasia

After the experiment, all rabbits were sacrificed by injecting 10 mL of air through the ear-rim auricular vein.

#### 2.2.5. Gross Specimen

The rabbit liver was harvested, and the ablation lesion was split along the ablation needle path. The length and width diameters of the lesion were measured.

#### 2.2.6. Statistical Analysis

All statistical analyses were performed using the SPSS 19.0 computer software package (SPSS, Inc., Chicago, IL, USA). A value of *p* < 0.05 was considered to indicate a statistically significant difference between the groups. All data are presented as mean ± SD. For the counting variables, a *t*-test, or Mann–Whitney U-rank test was used to compare the two groups. The consistency of the efficacy evaluation was analyzed using the Bland–Altman test. Young’s modulus under different ablation conditions before and after ablation was used in the factorial analysis.

## 3. Results

### 3.1. The Situation of the Ablation Zone

All twenty livers of New Zealand rabbits successfully formed the ablation zone after RFA. The vaporization area could be seen during the ablation process. The distance from the needle to the liver surface is about 1 cm.

### 3.2. RTE for the Detection of the Ablation Zone

The RTE images showed that the ablation zone had a clear boundary, and was indicated by a homogeneous blue, while the surrounding normal liver tissue was indicated by yellow, green, and red colors (Figure 3). For RFA, the ratio of the elastic value of the ablation zone rim to the surrounding normal liver tissue was 3.41 ± 0.67 (2.23–4.76). The area of the ablation zone measured by RTE was 1.089 ± 0.199 cm^2^ for RFA.

### 3.3. SWE for the Detection of the Ablation Zone

From SWE, we can also see the ablation zone had a clear boundary. The ablation zone showed a yellow–green–red tricolor change, and the surrounding normal liver tissue showed a uniform blue color. A 2 mm sampling frame was used to measure the elasticity of the ablation zone from the center to the periphery, and the changes in Young’s modulus values were analyzed. The Young’s modulus gradually decreases from the center of the ablation zone to the periphery (Figure 3). The Young’s modulus within the RFA ablation zone ranged widely from 33.2 to 70.4 kPa (50.7 ± 11.3 kPa), while the Young’s modulus at the edge of the ablation zone ranged from 25.1 to 36.3 kPa (28.5 ± 3.13 kPa) which was relatively stable. The area of the ablation zone was 1.059 ± 0.201 cm^2^ for RFA based on elastic images.

### 3.4. Contrast-Enhanced Ultrasound Examination

After the elastic examination, a CEUS examination was performed for the ablation zones of New Zealand rabbits. It indicated no enhancement during the entire period (Figure 3). The surrounding normal liver tissue indicated uniform iso-enhancement performance. The average area of the ablation zone was 3.091 ± 0.591 cm^2^ for RFA.

### 3.5. Gross Specimen Observation and Comparison of the Three Imaging Methods

From the gross specimens, we could see the ablation zone showed oval and indicated a uniform white change which meant necrosis (Figure 3). The area of the ablation zone was 1.081 ± 0.201 cm^2^ for RFA in the gross specimen. There was no significant difference in the area of the ablation zone, comparing RTE and SWE with the gross specimen (*p* = 0.791, 0.622) (Figure 4). There was also no significant difference between RTE and SWE (*p* = 0.351) (Figure 4). The Bland–Altman test showed that, as can be seen from Figure 5, the mean of the difference between the paired data of twenty cases using RTE and SWE was 0.03 cm^2^, and the 95% agreement margin was −0.03 to 0.09, and 10% (2/20) points are outside the 95% consistency limit. However, comparing RTE and SWE with CEUS, the CEUS measured a larger area (*p* = 0.000, 0.000) (Figure 4).

### 3.6. Comparison of Two Elastic Imaging Methods for the Detection of the Edge of the Ablation Lesions

Both RTE and SWE can show the contours of the ablation zone and define the edge. RTE can only semi-quantitatively define the edge of the ablation zone. In addition to that, SWE can show the edge of the lesion, and SWE can quantitatively indicate the fixed elasticity of the edge. The range of the ablation zone will change according to different scaling ranges with SWE. In our study, we use the measuring range scale of 90 kpa. There is no significant difference between the extent of the SWE and the gross specimen (*p* = 0.816).

## 4. Discussion

Our results demonstrated that RTE and SWE could effectively display the shape of the ablation zone of rabbit liver just after the thermal ablation. Moreover, both RTE and SWE could calculate the elastography value of the thermal ablation zone. There were no significant differences in the ablation area between RTE, SWE, and the gross specimen. However, CEUS area measurements were significantly larger than those measured by RTE, SWE, and the gross specimen.

Real-time ultrasound imaging is widely used to guide the RFA treatment, and it could provide the visualization of the ablation zone [14,15]. However, the gas bubbles that formed during tissue heating hinder the evaluation of the ablation boundary. Although after the ablation we could see a different echogenic zone, obviously it is not accurate. It is also difficult to judge whether there is incomplete ablation [16]. During CEUS, microbubble-specific software on the sonographic console weakens all background signal intensity so that the operator can clearly see the signal intensity produced by the contrast agent passing under the sonographic probe, while the nonvascularized (unenhanced) tissue remains invisible [17]. CEUS has been used to evaluate the ablation effect in many studies [18,19]. However, due to the presence of vaporization mass after ablation and the hyperemia reaction around the ablation zone, it is difficult to accurately determine the extent of the lesion and evaluate the ablation effect in the short-term after ablation using contrast-enhanced ultrasound. Nishigaki et al. [20] found that the outline of the coagulated tumor became gradually more obscure with time, so they assessed the ablative margin at 3 h after the RFA procedure. The situation of the lesion cannot be effectively observed in time, and it is impossible to know whether or not the complete ablation is achieved, which brings difficulties for subsequent treatment. Contrast-enhanced computed tomography has been the gold standard for predicting the region of coagulation necrosis, but ionizing radiation is a big challenge for patients [21,22]. Moreover, it is not real-time. Elastographic imaging techniques provide an alternative for monitoring ablation extent that can be used in real time on the imaging system [23], already being utilized for needle guidance without the need for contrast agents. RTE and SWE are the most commonly used elastographic imaging methods [13,24]. RTE uses elastic maps to reflect tissue strain or measure the elastic ratio of lesions to surrounding tissue as a qualitative or semi-quantitative measure. SWE is a quantitative measurement method that by measuring the shear wave velocity and Young’s modulus value, respectively, reflects the hardness of the organization. At present, the two methods are widely used in clinical settings, for the thyroid, breast, and liver lesions in different parts of the differential diagnosis.

After radiofrequency ablation, coagulation necrosis occurs, and the lesion becomes harder than the surrounding normal liver tissue. It is possible to judge the boundary of the lesion based on the function of elastography to evaluate the hardness [25,26,27]. This becomes the theoretical basis for our use of elastography to detect lesions. Xu et al. [28] used SWE to measure the elasticity of RFA lesions and indicated it could qualitatively and quantitatively measure the ablation zones. Tian et al. [29] conducted research on measuring the elasticity of the extracorporeal liver after ablation, which indicated that the SWE measurement of hardness was not affected by the gas, and it could indicate the region of the ablation zone clearly. The gasification mass of the ablation zone has less effect on the elasticity. It is possible to judge the situation of the lesion in real time after the ablation in the short term. In our study, both the SWE and RTE could indicate the edge of the ablation zone, while it is difficult to manifest the ablation zone clearly with 2D ultrasound. There was no significant difference between the diameter of the lesion and the gross specimen measured by elastic ultrasound. SWE can show the elastic Young’s modulus value, which has certain quantitative significance for the judgment of the edge of the lesion [23,30]. The display range of CEUS is more extensive than elastic and gross specimens. The reason is that the rim of the ablation zone is an ischemic area, so the regional tissue did not suffer complete necrosis, but the local blood supply disappeared. However, the extent of necrosis in actual lesions is limited.

This study has some limitations. First, this study only tested the lesions of normal liver tissue with ablation, while we did not ablate the transplanted tumors. Because the study focused on the elastotic measurement of the ablation zone, the conclusion could support our opinions. Second, the study only measured the elastotic change within half an hour, but we did not make the measurement for the different time periods after the operation. Thus, in the future, we will measure the elastotic change during different time periods. Third, elastographic evaluation and CEUS examination were not performed on the liver of each rabbit prior to ablation.

In conclusion, RTE and SWE are both able to confirm accurately the range of ablation zones shortly after the ablation for rabbit livers. The quantitative measurement of ablation edges by SWE provides guidance for the future determination of ablation margins in clinical applications.

## Figures and Tables

**Figure 1 diagnostics-13-01145-f001:**
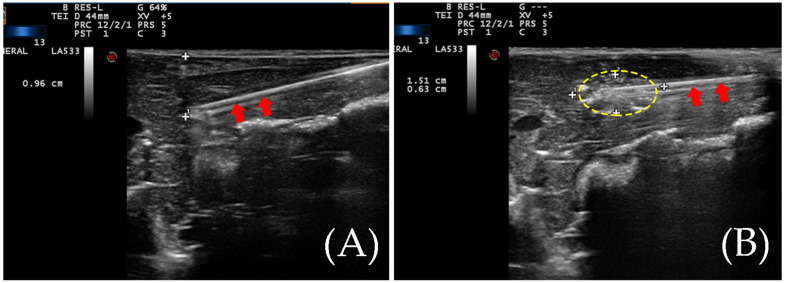
(**A**) An ablation needle was inserted during RFA. The tip of the ablation needle is about one centimeter from the liver surface. The red arrows indicate the ablation needle rod. (**B**) The yellow dotted circle indicates the vaporization zone during the ablation.

**Figure 2 diagnostics-13-01145-f002:**
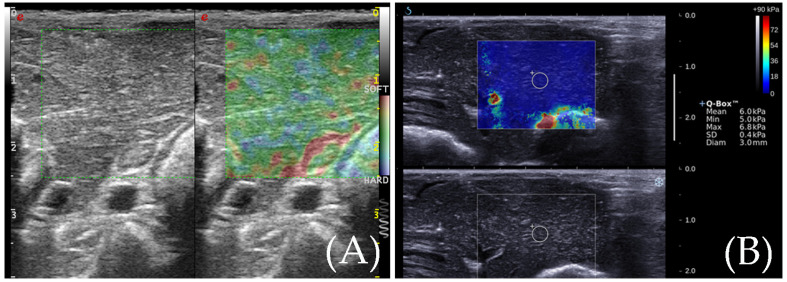
(**A**) RTE elasticity image of normal rabbit liver and the normal rabbit liver elasticity diagram show three colors: yellow, green, and red. (**B**) SWE elasticity image of normal rabbit liver and the normal rabbit liver elasticity diagram show an even blue color.

**Figure 3 diagnostics-13-01145-f003:**
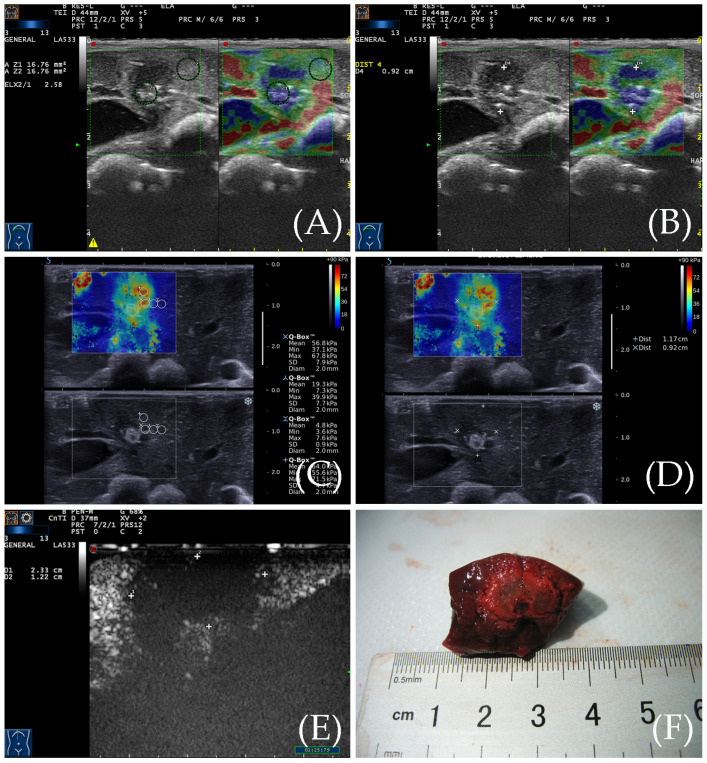
(**A**) RTE elasticity image of the ablation zone. The even blue indicates the ablation zone, and its boundary is clear. The yellow, green, and red indicate the peripheral normal liver. (**B**) The diameter of the ablation zone was measured according to the elasticity image. (**C**) SWE elasticity image of the ablation zone. The ablation zone is indicated by three colors: yellow, green, and red. The peripheral normal liver is indicated by an even blue color. (**D**) The diameter of the ablation zone was measured according to the elasticity image. (**E**) After CEUS, the diameter of the ablation zone was measured by CEUS image. (**F**) The gross specimen of the ablation zone was measured after sacrifice.

**Figure 4 diagnostics-13-01145-f004:**
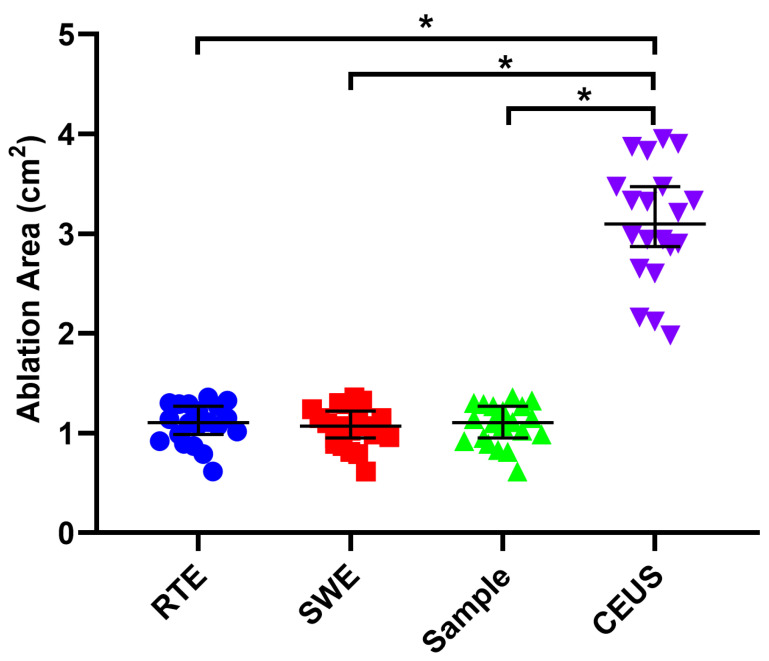
The ablation area was measured by different measurement methods. There was no statistical difference in the area of the ablation lesions measured by RTE, SWE, and gross specimen. The area of ablation lesions measured by CEUS is larger than the first three, and their differences are statistically significant. Results are represented as the means ± SD. Blue dots mean RTE group, red dots mean SWE group, green dots mean Sample group, purple dots mean CEUS group, * *p* < 0.05.

**Figure 5 diagnostics-13-01145-f005:**
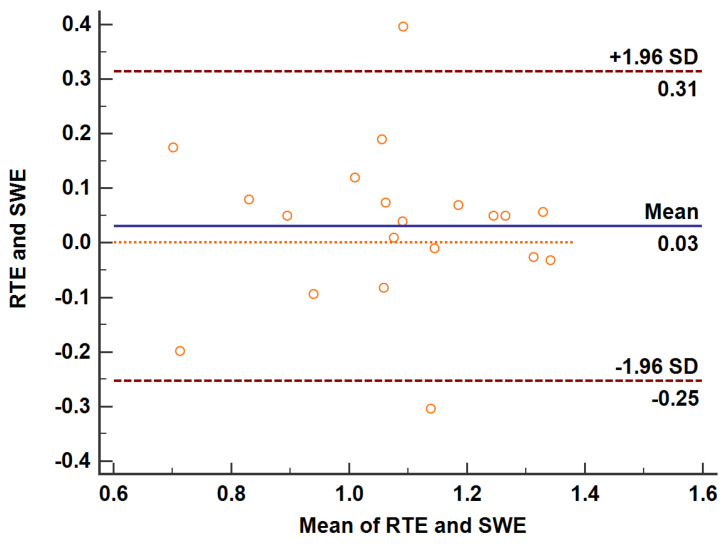
Bland–Altman map of the differences between RTE and SWE for the ablation zone after treatment. The Bland–Altman test was used to measure the ablation area after treatment. The mean of the difference between the paired data of twenty cases was 0.03 cm^2^, the 95% agreement margin was −0.03 to 0.09, and 10% (2/20) points are outside the 95% consistency limit. RTE, real-time elastography; SWE, shear wave elastography.

## Data Availability

Data sharing not applicable.

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
