# Peer review of "Real-Time Elastography versus Shear Wave Elastography on Evaluating the Timely Radiofrequency Ablation Effect of Rabbit Liver: A Preliminary Experimental Study"

_diagnostics, 2023, doi:10.3390/diagnostics13061145_

Round 1

Reviewer 1 Report

The study aims to evaluate the timely radiofrequency ablation effect of rabbit livers using RTE and SWE, and compare the ablation areas which were calculated by RTE, SWE, CEUS and gross specimens.

It is an interesting study, with practical applicability, but certain aspects must be discussed more.

1. the major limitation of the study is the absence of elastographic evaluation and CEUS before ablation. As a result, it is not possible to know what the appearance of the respective areas was before the ablation. It was possible that those areas had changes even before the ablation. Even if this was unlikely, this aspect decreases the rigor of the study. Please insist on this aspect in the discussion chapter

2. how was the health status of the animals included in the study documented?

3. Elastographic evaluation through RTE: the authors specify: “When the images are stable, the operator freezes the image and selects the satisfied image for analyzing. The elastic ratio of ablation area and normal liver tissue was measured for three times and took the average of the measured value". Why were the measurements performed on the same image and not on different images? The result would have been much more credible if the process of obtaining the stable image had been repeated 3 times and therefore the measurements would have been made on different images

4. Elastographic evaluation by SWE (Supersonic): did you use a quantitative image stability index (eg SI>90%)) or only a qualitative one? Please comment

5. If the size of the modified area after ablation was larger than the ROI and, possibly, inhomogeneous, where exactly was the ROI placed for the elastographic evaluation? The result could have been different depending on the area where the ROI is placed.

6.  the article needs English language corrections

Reviewer 2 Report

The authors showed in an animal experiment that RTE and SWE are accurate to analyze the extent of the ablation zone on liver shortly after the procedure. Please find my comments below:
1. The study looks promising, however, the English language is very difficult to understand.

2. The abbreviations in the abstract shouldn’t be used because they make the text even more difficult to understand. My suggestion is to revise the English language of the paper above all.

3. Secondly, please explain or reference the method of SWE/RTA analysis. Do you manually measure the velocity of the shear waves, or is there an automatic tool ?

Round 2

Reviewer 1 Report

The authors responded to all my observations. I think the article can be published in this form.

Reviewer 2 Report

Thank you for the revision of the paper.